# Association between the Processed Dietary Pattern and Tumor Staging in Patients Newly Diagnosed with Head and Neck Squamous Cell Carcinoma

**DOI:** 10.3390/cancers15051476

**Published:** 2023-02-25

**Authors:** Ana Carolina da Silva Lima, Tathiany Jéssica Ferreira, Adriana Divina de Souza Campos, Larissa Morinaga Matida, Maria Beatriz Trindade Castro, Ana Amélia Freitas-Vilela, Maria Aderuza Horst

**Affiliations:** 1Nutritional Genomics Research Group, Faculty of Nutrition, Federal University of Goiás, Goiânia 74605-080, Brazil; 2Josué de Castro Nutrition Institute, Federal University of Rio de Janeiro, Rio de Janeiro 21941-901, Brazil; 3Special Academic Unit of Health Sciences, Federal University of Jataí, Jataí 75801-615, Brazil

**Keywords:** food, diet, head and neck neoplasms, tumor stage, prognosis

## Abstract

**Simple Summary:**

Most patients with head and neck squamous cell carcinoma (HNSCC) are diagnosed in the advanced stages of the disease. The tumor stage is one of the most important prognostic factors for patients with this type of cancer. However, it is unknown whether the dietary pattern is related to prognostic factors in cancer, such as the tumor stage and cell differentiation. The aim of our cross-sectional study was to investigate the association between dietary patterns, tumor staging, and the degree of cellular differentiation in patients newly diagnosed with HNSCC. We found that a greater adherence to a dietary pattern consisting of processed foods was associated with advanced staging. This information contributes to the construction of nutritional guidance for reducing the risk of HNSCC.

**Abstract:**

Purpose: This study aimed to assess the association between dietary patterns and tumor staging and the degree of cell differentiation in patients with head and neck squamous cell carcinoma (HNSCC). Methods: This cross-sectional study included 136 individuals newly diagnosed with different stages of HNSCC, aged 20- to 80 years-old. Dietary patterns were determined by principal component analysis (PCA), using data collected from a food frequency questionnaire (FFQ). Anthropometric, lifestyle, and clinicopathological data were collected from patients’ medical records. Disease staging was categorized as initial stage (stages I and II), intermediary (stage III), and advanced (stage IV). Cell differentiation was categorized as poor, moderate, or well-differentiated. The association of dietary patterns with tumor staging and cell differentiation was evaluated using multinomial logistic regression models and adjusted for potential confounders. Results: Three dietary patterns, “healthy,” “processed,” and “mixed,” were identified. The “processed” dietary pattern was associated with intermediary (odds ratio (OR) 2.47; 95% confidence interval (CI) 1.43–4.26; *p* = 0.001) and advanced (OR 1.78; 95% CI 1.12–2.84; *p* = 0.015) staging. No association was found between dietary patterns and cell differentiation. Conclusion: A high adherence to dietary patterns based on processed foods is associated with advanced tumor staging in patients newly diagnosed with HNSCC.

## 1. Introduction

Head and neck squamous cell carcinoma (HNSCC) is a heterogeneous disease that includes epithelial malignancies localized to the oral cavity, pharynx, larynx, nasal cavity, and paranasal sinuses [1]. Worldwide, it accounts for approximately 4.5% of all cancer cases. HNSCC is responsible for approximately 878,000 new cases per year, causing 444,000 deaths in 2020 [2,3]. In addition, HNSCC can be asymptomatic for a long period, and patients are often diagnosed at advanced stages of the disease (III and IV) and in metastatic stages, which leads to a worse prognosis and a reduced cure rate [1,4,5].

The tumor stage is one of the most important prognostic factors for patients with HNSCC and is commonly described by the TNM staging system [1,6]. Primary tumor extension (T), lymph node metastasis (N), and distant metastasis (M) set the TNM classification of malignant tumors as globally recognized standards for classifying the extent of cancer spread [7,8]. Determining the pathological degree of tumor differentiation is useful because poorly differentiated neoplasms often have a worse prognosis than well-differentiated tumors [9]. The prognosis of head and neck cancer depends on several factors, such as the tumor stage, the degree of cellular differentiation [6], and lifestyle factors, such as smoking, alcohol consumption [10,11], body mass index (BMI) [12,13], and diet [14,15].

In addition, the most recognized environmental risk factors for HNSCC are tobacco, alcohol, and human papillomavirus (HPV) infection in oropharyngeal cancer [10,16]. However, evidence shows that dietary patterns can also contribute to its pathogenesis [17,18,19,20]. A population-based case-control study (2002–2006) included 1176 HNSCC cases and 1317 age-, race-, and sex-matched controls from central and eastern North Carolina. The authors found that a dietary pattern based on fruits, vegetables, and lean meat was associated with a reduced risk of HNSCC, whereas a dietary pattern based on fried processed meats, high in fats and sweets, may increase the likelihood of developing cancer [19]. Therefore, food and dietary patterns have been described as modulators of the risk of developing disease and mortality in HNSCC [14,18,19,20]. However, little is known about the influence of diet on prognostic factors in HNSCC.

To our knowledge, no study has evaluated the relationship between dietary patterns and prognostic factors, such as tumor staging and cellular differentiation, in HNSCC. Therefore, our study is the first to investigate the association between dietary patterns, tumor staging, and the degree of cellular differentiation in patients newly diagnosed with HNSCC.

## 2. Materials and Methods

### 2.1. Design and Patients

This cross-sectional study was conducted at the Hospital Araújo Jorge (HAJ), Goiânia, Brazil, from January 2017 to February 2018. One hundred and thirty-six patients between 20 and 80 years of age, comprising both sexes, with a recent diagnosis of HNSCC but still in the pre-treatment phase, were recruited. Patients affected by lip cancer, patients diagnosed with a second primary tumor regardless of the site (including non-melanoma skin), relapse, and severe comorbidities, such as autoimmune diseases, or patients who acquired immunodeficiency syndrome, chronic renal failure, and renal failure were not included. This study was conducted in accordance with the Declaration of Helsinki, and ethical approval was provided by the Research Ethics Committee of the Federal University of Goiás, the National Human Research Ethics Committee of the Brazilian Ministry of Health (CONEP), and the Ethical Review Committee of the HAJ. All patients provided signed, free, and informed consent forms before participation.

### 2.2. Dietary Data

After obtaining informed consent, dietary data were collected using a semiquantitative food frequency questionnaire (FFQ) previously used in the Brazilian Longitudinal Study of Adult Health (ELSA-Brazil) and validated by Molina et al. [21]. This FFQ has three components: (1) foods/preparations, (2) the measurement of portion intake, and (3) the frequency of consumption, with eight response options converted into daily frequency intake.

To estimate the usual consumption in the last 12 months, the participants were asked about their intake of foods listed on the FFQ and the frequency of consumption in a day, week, or month. Afterwards, the food consumption frequency was recorded as daily numerical quantitative variables: never or rarely = 0; 1 to 3×/month = 0.067; 1×/week = 0.143; 2 to 4×/month = 0.429; 5 to 6×/week = 0.786; 1×/day = 1; 2×/day = 2; and 3× or more/day = 3.5.

The FFQ list contained 74 food items, including nonalcoholic and alcoholic beverages. Some foods, such as “acarajé” and “chimarrão,” were excluded from the list because they are not a part of the region’s typical diet, and the frequency of their consumption is low. The remaining 72 items were combined into 19 food groups based on similarities in nutrient composition and consumption frequency. Items consumed by more than 80%, such as rice, beans, bread, and coffee [22], or those that presented differences in nutritional composition, such as fish, were grouped separately.

The 19 food groups were as follows: (i) rice: white and brown rice; (ii) grains and oilseeds: oats, granola, grain bran and other cereals, nuts, cashew nuts, Brazil nuts, peanuts, almonds, and pistachios; (iii) tubers, roots, cereals, and legumes: yellow potatoes, cassava, corn, vegetable soup, and lentils; (iv) pasta and flour: cassava flour, farofa, polenta, and pasta; (v) fast foods: pizza, snacks, cheese bread, ice cream, chocolate, and pudding; (vi) beans; (vii) breads: sweet bread, bread, loaf of bread, whole wheat bread, light bread, and light whole meal bread; (viii) cakes and cookies: cake without filling, sweet biscuit with filling, sweet biscuit without filling, and salty biscuit; (ix) milk and dairy products: semi-skimmed milk, skimmed milk, whole milk, soy milk, white cheese, yellow cheese, yogurt, and light yogurt; (x) vegetables and legumes: lettuce, kale, cabbage, chicory, tomato, squash, zucchini, green beans, okra, cauliflower, broccoli, carrots, and beets; (xi) fruits: orange, banana, papaya, apple, watermelon, melon, pineapple, mango, grape, and fresh juice; (xii) meat and eggs: beef, chicken breast, chicken (other parts), and egg; (xiii) pork and sausages: pork, tripe, ham, and sausage; (xiv) fish: boiled fish and fried fish; (xv) fats: margarine and butter; (xvi) sugar-sweetened beverages: soda, diet/light soda, powdered juice, and industrial juice; (xvii) coffee; (xviii) beer and wines: beer, red wine, and white wine; (xix) distilled beverages: cachaça, whiskey, and vodka.

### 2.3. Clinical Staging and Cell Differentiation

The clinical staging of the tumor was performed by a head and neck surgeon based on imaging (computed tomography, magnetic resonance imaging, and/or videolaryngoscopy) and anatomopathological findings. Tumor staging was based on the TNM classification, as recommended by the American Joint Committee on Cancer (AJCC)/Union for International Cancer Control (UICC) in the TNM Classification of Malignant Tumours (VII edition). These data were collected from the patients’ charts, who were then categorized based on stages (I–IV stage) [7]. In this study, the clinical stages of the disease were grouped as initial (stages I and II), intermediary (stage III), and advanced (stage IV). The degree of cell differentiation was obtained from medical charts, and the tumors were classified as well, moderately, or poorly differentiated.

### 2.4. Variables

Data, such as age (years), sex (male and female), smoking status (current/former or never), drinking status (current/former or never), primary anatomic site, and height (m), were collected from medical records and face-to-face interviews. Data on smoking and drinking status categorized as current were based on the 12 months prior to cancer diagnosis. The primary anatomical site of the tumor was categorized into three groups: (1) the oral cavity, (2) the oropharynx, and (3) the larynx. To calculate the BMI (kg/m²), patient weight was obtained using a digital scale with a maximum capacity of 150 kg and a sensitivity of 100 g (Tanita^®^, Arlington Heights, IL, USA), and height was obtained from medical records; BMI was calculated by dividing the weight (kg) by the height (m) squared. Based on the BMI, the nutritional status of each participant was determined according to age: (i) in adults: underweight (<18.5 kg/m²), normal weight (18.5–24.9 kg/m²), overweight (25.0–29.9 kg/m²), and obesity (>30 kg/m²) [23]; (ii) in the elderly: malnutrition (<22 kg/m²), normal weight (22–27 kg/m²), and overweight (>27 kg/m²) [24].

### 2.5. Statistical Analysis

The power of the test was estimated a posteriori and was based on Pr(Y = 1|X = 1) H0 equal to 0.15 and Pr(Y = 1|X = 1) H1 equal to 0.25, for a sample of 136 individuals. The calculations showed that, at a significance level of 5%, the statistical power was equivalent to 83%. G-Power ^®^ software (version 3.1.9.2) was used.

Descriptive data are presented for all demographic, anthropometric, lifestyle, and clinicopathological variables in percentages (%). Principal component analysis (PCA) was used to determine the dietary patterns of the patients. A correlation matrix was constructed to assess the correlation between food groups. Bartlett and Kaiser–Meyer–Olkin (KMO) tests were used to assess the applicability of the factor method to the dataset, with KMO considered adequate when >0.60 [25]. Eigenvalues greater than 1.5 were adopted for data interpretation. Varimax rotation was applied to obtain the orthogonal factors. Food groups that had factor loadings >0.20 and communalities >0.20 were considered strongly associated with dietary patterns. Food patterns were named according to the following criteria: (A) the major food groups that made up each factor, (B) the highest factor loadings of each identified pattern, and (C) the interpretation of the food patterns.

The association between dietary patterns and tumor staging and cell differentiation was evaluated using multinomial logistic regression models. The dietary pattern was identified as the exposure variable, and tumor staging and cell differentiation were identified as the outcome variables, with the initial stage (stages I and II) and well-differentiated tumors as the basis for the outcome. The multinomial models included the potential confounders selected with directed acyclic graphs (DAGs), using the Dagitty software available online at http://dagitty.net/development/dags accessed on 3 September 2021. The DAG was built using variables considered clinically relevant based on biological mechanisms or evidence from previously published data. The confounding factors identified were sex (male or female), age (years), smoking status (current/former or never smoked), and alcoholism (current/former or never drank). The adjusted odds ratios (OR) of the variables in the final model are presented with their 95% confidence intervals (CI). Statistical significance was set at a *p*-value less than 0.05. Statistical analyses were performed using STATA software version 15.0.

## 3. Results

### 3.1. Sample

Among the 136 patients, most were male (78.7%), aged <60 years (55.9%; mean age, 59 years (standard deviation ± 10)), smokers (89.1%), and drinkers (83.2%). The most common anatomical site of the tumor was the oropharynx (37.5%), followed by the oral cavity (33.8%) and larynx (28.7%). More than 82% of the patients had the intermediate and advanced stages of the disease (III and IV), and 58% had moderately differentiated tumor cells (Table 1).

### 3.2. Dietary Patterns

Three dietary patterns were identified using PCA. The KMO test (0.600) (Appendix A) showed that the correlation between the food groups was sufficient and appropriate for PCA. The first pattern, termed “healthy,” was characterized by the consumption of grains, tubers, dairy products, vegetables, fruits, and fish. The second pattern, termed “processed,” was characterized by the consumption of fast food, bread, cake and cookies, meat and eggs, pork and sausages, fat, and sugar-sweetened beverages. The third pattern, termed “mixed,” was characterized by the consumption of rice, pasta, flour, beans, coffee, beer, and distilled beverages. The eigenvalues for each dietary pattern were 2.79 (healthy), 1.99 (processed), and 1.66 (mixed) (Appendix A). The percentage of variance explained in each pattern was 11.93, 11.50, and 10.57, respectively.

### 3.3. Association of Dietary Patterns with Tumor Staging and Degree of Cell Differentiation

The “processed” dietary pattern was associated with intermediate (III) and advanced (IV) tumor stages in the model without adjustment for covariates. After adjustment, the associations remained significant (*p* < 0.050) (Table 2). Individuals with a higher adherence to the “processed” pattern were 2.47 (odds ratio (OR) = 2.47; 95% confidence interval (CI) = 1.43–4.26; *p* = 0.001) and 1.78 (OR = 1.78; 95% CI = 1.12–2.84; *p* = 0.015) times more likely to have intermediate and advanced tumor stages, respectively, when compared to patients with the initial tumor stages (I and II). For the “healthy” and “mixed” dietary patterns, we did not find significant associations (*p* > 0.05). Further, dietary patterns were not associated with the degree of cell differentiation (*p* > 0.05) (Table 2).

## 4. Discussion

In our study, three dietary patterns, “healthy,” “processed,” and “mixed,” were identified in patients newly diagnosed with HNSCC in the pre-treatment phase. The “processed” dietary pattern was associated with advanced tumor stages (III and IV) when compared to patients with initial tumor stages. Although some studies have evaluated dietary patterns in the HNSCC population [17,18,19,20], to the best of our knowledge, this is the first study to investigate the relationship between dietary patterns and prognostic factors, including tumor staging and the degree of cell differentiation, in pre-treatment HNSCC patients. These results warrant interest because advanced-stage tumors have a worse prognosis, with a high risk of relapse and metastasis [5].

Crowder et al. [17] assessed food intake according to dietary patterns in the head and neck cancer pre-treatment population, and two predominant patterns were observed. The first one was named a “prudent” dietary pattern, characterized by a high intake of fruit, vegetables, whole grains, low-fat dairy, legumes, and less saturated fat. The second was the “Western” pattern, which included a high intake of red and processed meats, refined grains, potatoes, French fries, high-fat dairy, condiments, desserts, snacks, and sugar-sweetened beverages. Similarly, Arthur et al. [18] assessed dietary patterns in patients with newly diagnosed HNSCC and found two patterns. The first pattern was “whole foods,” characterized by a high intake of healthy foods, including vegetables, fruits, legumes, fish, whole grains, and others. The second pattern, named “Western,” involved the consumption of foods prevalent in the Western diet, such as a high intake of red and processed meats, refined grains, fried foods, and high-fat and ultra-processed foods. Furthermore, the study also showed that pre-treatment HNSCC patients with a higher adherence to the “whole foods” pattern had a lower risk of relapse and longer survival [18]. Our results showed dietary patterns that were consistent with those described in the literature. The “healthy” pattern is marked by a predominance of fruits, vegetables, tubers, grains, and fish, and the “processed” pattern is characterized by the consumption of sugary, fatty, and ultra-processed foods.

There is evidence that a dietary pattern characterized by the consumption of fruits, vegetables, and lean meat is associated with a reduced risk of HNSCC [19]. In addition, a low fruit intake is negatively associated with survival in patients with this disease [26]. The protective effects of fruits and vegetables could be explained by the fact that they are dietary sources of bioactive compounds in food, such as polyphenols and carotenoids. These compounds play an antitumor role by reducing the risk of DNA damage. They can suppress the expression of oncogenes, activate the expression of tumor suppressor genes, and induce apoptosis and cell differentiation. Additionally, these compounds can modulate angiogenesis and immune responses in individuals [27,28]. Therefore, one of the hypotheses tested in our study was the association of “healthy” dietary patterns with initial staging. However, no such association was found. This result may be explained by the fact that most patients with HNSCC are diagnosed in more advanced stages of the disease [1,4,5], and the intake of a healthier dietary pattern may have started after the diagnosis of HNSCC.

Bradshaw et al. [19] found that a dietary pattern consisting of fried foods, processed and high-fat meats, and sweets was associated with an increased risk of HNSCC, especially laryngeal tumors. Our results are in agreement with those reported by Bradshaw et al. [19] who reported a dietary pattern characterized by the consumption of sugary, fatty, and ultra-processed foods. Arthur et al. [18] observed that high intakes of total carbohydrates, total sugar, glycemic load, and simple carbohydrates were associated with an increased risk of all-cause mortality compared with low intakes. Sugars are the main energy source for tumor cells, which are glucose-dependent for growth and proliferation [29]. Several mechanisms support the hypothesis of a relationship between high sugar consumption and an increased risk of cancer. These mechanisms include adiposity, the disruption of insulin signaling pathways, inflammation, oxidative stress, altered gene expression, and DNA damage, resulting in changes in cell proliferation and differentiation as well as the inhibition of apoptosis [29].

Additionally, another explanation could be that the intake of high-glycemic-load foods promotes increased levels of serum glucose, which promotes a compensatory increase in insulin levels [15]. Insulin, in turn, is an anabolic hormone that accelerates glucose uptake and stimulates mitosis, which can promote tumor cell proliferation [19,30]. In this manner, our results provide evidence that the consumption of foods with a high energy density and low nutritional value may be associated with a worse HNSCC prognosis.

Some foods belonging to the “processed” dietary pattern, such as ham and sausage, contain artificial dyes known for their carcinogenic potential [31]. Moreover, other compounds, such as acrylamide and acrolein, can be formed during thermal processing. These compounds have been linked to an increased risk of endometrial and kidney cancers in non-smokers [32]. Furthermore, a greater consumption of ultra-processed foods is associated with greater oxidative DNA damage, which may promote cancer initiation and development [32]. Our results are consistent with those reported in the literature and with the possible oncogenic mechanisms involved in the process. However, further studies are required to investigate the mechanisms that may be related to HNSCC.

It is well documented in the literature that alcohol consumption is one of the main risk factors for HNSCC and is associated with a worse prognosis [6,10,16]. We hypothesized that we would find a dietary pattern mainly composed of alcoholic beverages and other unhealthy foods associated with advanced stages of the disease. However, we did not find an association between the “mixed” pattern and prognostic factors. This result may be explained by the presence of other foods with balanced nutritional composition, such as rice and beans.

As strengths of this study, we highlight the use of a validated FFQ for chronic diseases and the researchers’ previous training. However, this instrument is limited by the participant’s memory bias in reporting the frequency and amount of food. Other study limitations include the cross-sectional design, which does not allow for the determination of causality, the number of patients diagnosed in the early stages of the disease, which was lower than that in patients in an advanced stage, and the impossibility of assessing the HPV status, as this is an important etiologic factor in oropharyngeal cancer. However, HPV-induced HNSCC tends to occur more often in non-smokers, unlike our study, in which the population is mostly smokers. Therefore, a lack of information on HPV status may not affect the outcomes in this patient population.

## 5. Conclusions

In conclusion, a high adherence to the dietary pattern described as “processed” was associated with a higher risk of intermediate and advanced tumor stages in patients newly diagnosed with HNSCC. Our findings suggest that the consumption of processed foods rich in sugars and fats may contribute to the progression of HNSCC. However, further studies should be conducted to investigate the relationship between dietary patterns and tumor stages, cell differentiation, and associated mechanisms. In addition, future research should evaluate dietary patterns before and after disease treatment, as well as other indicators, such as prognosis and survival.

## Figures and Tables

**Table 1 cancers-15-01476-t001:** Clinical and pathological data of patients with HNSCC (*n* = 136).

Description of the Patients	N (%)
Age (years)	
<60	76 (55.9)
≥60	60 (44.1)
Sex	
Male	107 (78.7)
Female	29 (21.3)
Skin color	
White	45 (33.1)
Brown	77 (56.6)
Black	14 (10.3)
BMI (kg/m2)	
Adults (*n* = 76)	
Underweight (<18.5)	22 (28.9)
Normal weight (18.5–24.9)	25 (32.9)
Overweight (25–29.9)	16 (21.1)
Obesity (>30)	6 (7.9)
NA	7 (9.2)
Elderly (*n* = 60)	
Malnutrition	27 (45.0)
Normal weight	19 (31.7)
Overweight	10 (16.7)
NA	4 (6.6)
Smoking status	
Current/former	122 (89.7)
Never	14 (10.3)
Drinking status	
Current/former	114 (83.8)
Never	21 (15.4)
NA	1 (0.8)
Anatomic site	
Oral cavity	46 (33.8)
Oropharynx	51 (37.5)
Larynx	39 (28.7)
Tumor staging	
I–II (initial)	22 (16.2)
III (intermediary)	23 (16.9)
IV (advanced)	89 (65.4)
NA	2 (1.5)
Cell differentiation	
Well-differentiated	18 (13.2)
Moderately differentiated	79 (58.1)
Poorly differentiated	26 (19.1)
NA	13 (9.6)

BMI, body mass index; HNSCC, head and neck squamous cell carcinoma; NA, data unavailable.

**Table 2 cancers-15-01476-t002:** Multinomial logistic regression and 95% confidence intervals (95% CI) for cell staging and differentiation and dietary patterns in pre-treatment HNSCC patients.

	Dietary Patterns					
Variables	Healthy OR (95% CI)	*p*-Value	Processed OR (95% CI)	*p*-Value	Mixed OR (95% CI)	*p*-Value
	Unadjusted					
Staging						
Initial (I/II)			(Base outcome)			
Intermediary (III)	1.08 (0.70–1.69)	0.722	1.98 (1.25–3.15)	**0.004**	1.29 (0.81–2.06)	0.287
Advanced (IV)	1.34 (0.93–1.91)	0.112	1.54 (1.04–2.28)	**0.031**	1.43 (0.97–2.12)	0.074
Cell differentiation						
Well-differentiated			(Base outcome)			
Moderately differentiated	0.96 (0.69–1.35)	0.829	1.05 (0.73–1.51)	0.797	0.95 (0.67–1.36)	0.788
Poorly differentiated	0.91 (0.61–1.37)	0.655	1.00 (0.65–1.54)	0.986	1.09 (0.73–1.64)	0.659
	Adjusted					
Tumor staging						
Initial (I/II)			(Base outcome)			
Intermediary (III)	1.20 (0.73–1.97)	0.479	2.47 (1.43–4.26)	**0.001**	1.16 (0.71–1.88)	0.547
Advanced (IV)	1.48 (0.97–2.24)	0.067	1.78 (1.12–2.84)	**0.015**	1.21 (0.81–1.81)	0.345
Cell differentiation						
Well-differentiated			(Base outcome)			
Moderately differentiated	1.00 (0.69–1.45)	0.983	1.06 (0.71–1.59)	0.775	0.87 (0.60–1.28)	0.495
Poorly differentiated	0.96 (0.62–1.49)	0.846	0.99 (0.63–1.58)	0.985	0.87 (0.56–1.36)	0.553

Adjusted for sex, age, smoking, and alcohol status. OR, odds ratio; CI, confidence interval.

## Data Availability

The data presented in this study are available on request from the corresponding author (M.A.H.).

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
