# Peer review of "Association between the Processed Dietary Pattern and Tumor Staging in Patients Newly Diagnosed with Head and Neck Squamous Cell Carcinoma"

_cancers, 2023, doi:10.3390/cancers15051476_

Round 1

Reviewer 1 Report

I thank the authors to read this article titled "Association Between Processed Dietary Pattern and Tumor  Staging in Patients Newly Diagnosed with Head and Neck  Squamous Cell Carcinoma"
The limited number of patients is a great bias.

Author Response

Dear reviewer,

The authors would like to thank you for the relevant comments and suggestions, which contribute to improving the quality of the manuscript entitled “Association Between Processed Dietary Pattern and Tumor Staging in Patients Newly Diagnosed with Head and Neck Squamous Cell Carcinoma”. After meticulously analyzing the comments, questions and suggestions that were sent to us, we modified the manuscript in order to improve the overall understanding of our paper. We have responded to each of the comments below and included the track changes in the manuscript. We hope that this current version meets the expectations for the manuscript to be approved for publication in Cancers.

Comments to the Author:

1. I thank the authors to read this article titled "Association Between Processed Dietary Pattern and Tumor Staging in Patients Newly Diagnosed with Head and Neck Squamous Cell Carcinoma" The limited number of patients is a great bias.
A: We would like to thank you for carefully reading our manuscript and for the comment. We understand that the number of patients can be a limitation. However, we performed the calculation of the statistical power, which was greater than 80% (“The power of the test was estimated a posteriori and was based on Pr(Y=1|X=1) H0 equal to 0.15 and Pr(Y=1|X=1) H1 equal to 0.25, for a sample of 136 individuals. The calculations showed that at a significance level of 5%, the statistical power was equivalent to 83%. G-Power ® software (version 3.1.9.2)
was used” - Page 4 Line 156-159). Conventionally, the optimal power of a study is 0.8 (which can also be specified as 80%). DOI:
10.11613/BM.2021.010502

Sincerely

Reviewer 2 Report

This study aimed to assess the association between dietary patterns and tumor staging and the degree of cell differentiation in patients with head and neck cancer. This study is very interesting and bears clinical relevance. The design and analyses are also adequate. I only have some comments to the authors.

* In the 2.3. Clinical staging and cell differentiation (Page 3, Line 126-134): It is important to describe more details in this section. How were these patients staged? Moreover, it is also important to use CT or MRI during staging work-up. Without using these imaging studies, the stages described in this study may be concerned. 

* In the 2.3. Clinical staging and cell differentiation (Page 3, Line 126-134): Sarcoepnia is commonly seen in patients with head and neck cancer, and can exacerbate during their cancer treatment (e.g. radiotherapy) (J Cachexia Sarcopenia Muscle. 2020 Feb;11(1):145-159; Radiother Oncol. 2021 May;158:83-89).  If these imagings are available, I suggest the authors to analyze the association between dietary patterns and CT- or MRI-based muscle (i.e. sarcopenia). Otherwise, the authors may comment it in the Discussion section.

Author Response

Dear reviewer,

The authors would like to thank you for the relevant comments and suggestions, which contribute to improving the quality of the manuscript entitled “Association Between Processed Dietary Pattern and Tumor Staging in Patients Newly Diagnosed with Head and Neck Squamous Cell Carcinoma”. After meticulously analyzing the comments, questions and suggestions that were sent to us, we modified the manuscript in order to improve the overall understanding of our paper. We have responded to each of the comments below and included the track changes in the manuscript. We hope that this current version meets the expectations for the manuscript to be approved for publication in Cancers.

Comments to the Author:
1. In the 2.3. Clinical staging and cell differentiation (Page 3, Line 126-134): It is important to describe more details in this section. How were these patients staged? Moreover, it is also important to use CT or MRI during staging work-up. Without using these imaging studies, the stages described in this study may be concerned.

A: We would like to thank you for the comments. The contributions were fundamental for the improvement of the manuscript. TNM clinical staging data (T - Primary tumor extension; L - lymph node metastasis; M - distant metastasis) was performed by the head and neck surgeon. To determine the TNM staging of the patients, physical examination, imaging (computed tomography, magnetic resonance imaging and videolaryngoscopy) and anatomopathological
findings were considered. After the TNM classification, these were categorized into stages (I- IV stage) according to the recommendations of the American Joint Committee on Cancer (AJCC)/Union for International Cancer Control (UICC) TNM classification of malignant tumors (VII edition). These data were collected from the patients' charts.
We have added this information in the manuscript, 2.3 item, Page 3, Line 130-139.

2. In the 2.3. Clinical staging and cell differentiation (Page 3, Line 126-134): Sarcoepnia is commonly seen in patients with head and neck cancer, and can exacerbate during their cancer treatment (e.g. radiotherapy) (J Cachexia Sarcopenia Muscle. 2020 Feb;11(1):145-159; Radiother Oncol. 2021 May;158:83-89). If these imagings are available, I suggest the authors
to analyze the association between dietary patterns and CT- or MRI-based muscle (i.e. sarcopenia). Otherwise, the authors may comment it in the Discussion section.
A: Unfortunately, we do not have the images available to perform such an assessment. However, we will keep this suggestion in mind for the future studies.

Sincerely

Reviewer 3 Report

In this study the authors evaluated the correlation between nutrition forms and tumor stage and tumor differentiation among HNSCC patients. To this end, 136 patients were prospectively enrolled and had to fill out an "food frequency questionnaire" which has been established previously. Nutrition scores were divided into three domains and these domains were correlated to tumor stage and tumor diffentiation. Doing so, the authors found a significant correlation between so called "processed" dietary and intermediate/advanced tumor stage. The authors discuss these results in a thorough manner and try to explain why this correlation might be given the way it is. There are, however, some obstacles which hinder publication of the manuscript in the present form: Even though in the discussion section the authors name one main risk factor for HNSCC, namely alcohol intake, they miss to address the matter of smoking which is a even stronger risk factor for this tumor entity. Moreover, as far as I know there is a positive correlation between dietary, socioeconomical status, and smoking. Thus, it very well might be that due to the assumed lower socioeconomical status persons rather smoke AND live from a suboptimal dietary. Then again, this population might attend the hospital/doctor later. The authors should therefore, correlate smoking and dietary as well as smoking and tumor stage in a univariate and additionally in a multivariate manner. These results (if interesting) and the matter itself should be integrated into the discussion section. Since this study started in 2017 and with the 8th edition of TNM classification in force since January 2017, at least the p16 status of oropharyngeal cancers must be available. The role of HPV in oropharyngeal cancer should be mentioned in the introduction where risk factors for HNSCC are described. Perhaps even an HPV related subanalysis would be possible. 

Author Response

Dear reviewer,
The authors would like to thank you for the relevant comments and suggestions, which contribute to improving the quality of the manuscript entitled “Association Between Processed Dietary Pattern and Tumor Staging in Patients Newly Diagnosed with Head and Neck Squamous Cell Carcinoma”. After meticulously analyzing the comments, questions and suggestions that were sent to us, we modified the manuscript in order to improve the overall understanding of our paper. We have responded to each of the comments below and included
the track changes in the manuscript. We proofread the language (English) of the manuscript to improve language and grammar accuracy - review certificate was attached below. We hope that this current version meets the expectations for the manuscript to be approved for publication in Cancers.

Comments to the Author:
1. In this study the authors evaluated the correlation between nutrition forms and tumor stage and tumor differentiation among HNSCC patients. To this end, 136 patients were prospectively enrolled and had to fill out an "food frequency questionnaire" which has been established previously. Nutrition scores were divided into three domains and these domains were correlated to tumor stage and tumor diffentiation. Doing so, the authors found a significant correlation between so called "processed" dietary and intermediate/advanced tumor stage. The authors discuss these results in a thorough manner and try to explain why this correlation might be given the way it is. There are, however, some obstacles which hinder publication of the manuscript in the present form: Even though in the discussion section the authors name one main risk factor for HNSCC, namely alcohol intake, they miss to address the matter of smoking which is a even stronger risk factor for this tumor entity. Moreover, as far as I know there is a positive correlation between dietary, socioeconomical status, and smoking. Thus, it very well might be that due to the assumed lower socioeconomical status persons rather smoke AND live from a suboptimal dietary. Then again, this population might attend the hospital/doctor later. The authors should therefore, correlate smoking and dietary as well as smoking and tumor stage in a univariate and additionally in a multivariate manner. These results (if interesting) and the matter itself should be integrated into the discussion section. Since this study started in 2017 and with the 8th edition of TNM classification in force since January 2017, at least the p16 status of oropharyngeal cancers must be available. The role of HPV in oropharyngeal cancer should be mentioned in the introduction where risk factors for HNSCC are described. Perhaps even an HPV related subanalysis would be possible.

A: We would like to thank you for the comments. The contributions were fundamental for the improvement of the manuscript. We agree with your point, specifically for this manuscript we would like to clarify some points. The alcohol intake is mentioned in our discussion section as it is a dietary component of alcoholic beverages, which were assessed by the the food frequency questionnaire used in the present study. As tobacco use is one of the most important risk factors in HNSCC, smoking status was included as a covariate in the multinomial logistic regression model to minimize these biases. This information is described at Page 4 Line 179-181, and Table 2. Considering the importance of the role of HPV in the etiology of HNSCC, we have added this information in the introduction to Page 2, Line 60-61. We emphasize that one of the limitations of this study is the impossibility of assessing the status of HPV in the sample (Page 8, Line 300-303). Unfortunately, this analysis is not performed yet at Araujo Jorge hospital. Therefore, a sub analysis involving HPV is not possible to be performed.

Round 2

Reviewer 1 Report

Dear Authors,

thank you for answering my questions.
I think that the limited number is a bias, but it cannot be modified.

Author Response

Dear reviewer,

Thank you for your consideration. All questions were answered at the time of manuscript resubmission. The answers are attached. Please see the attachment.

Reviewer 2 Report

The authors had addressed my comments.

Author Response

(The authors gave the same response as above.)

Reviewer 3 Report

The authors did not meet the comments. For instance, the authors have to have p16 status of the patients since it is demanded for correct tumor staging according to the 8th edition of TNM classification. The authors did not respond to the issue of socioeconomic status which might go along with malnutrition and smoking. 

Author Response

Point 2: The authors did not meet the comments. For instance, the authors have to have p16 status of the patients since it is demanded for correct tumor staging according to the 8th edition of TNM classification. The authors did not respond to the issue of socioeconomic status which might go along with malnutrition and smoking.

Response 2: Dear reviewer, thank you for your comment, we would like to clarify some points. As mentioned in this manuscript, tumor staging was based on the TNM classification as recommended by the American Joint Committee on Cancer (AJCC)/Union for International Cancer Control (UICC) in the TNM Classification of Malignant Tumours (7th edition), which does not consider the status of HPV/p16. We updated the manuscript references, including the 7th edition (reference number 7) (Page 9 Line 353 – 354). Unfortunately, the p16 status data were not available at the Araujo Jorge hospital. Therefore, a subanalysis involving HPV is not possible to be performed. Despite the importance of HPV status, the authors believe that the lack of this data does not make the results found unfeasible. Other recent studies involving diet and head and neck cancer risk did not report data on HPV status were already published (DOI:  10.1002/ijc.31555; DOI: 10.1097/CEJ.0000000000000431; DOI: 10.1016/j.clnu.2019.03.030; DOI: 10.1097/EDE.0000000000000902; DOI:  10.18632/oncotarget.25288), including a study with the same patients as the present study (DOI: 10.1371/journal.pone.0220067).

In relation to socioeconomic status and their impact in malnutrition, the authors recognize the importance these factors. Therefore, the multinomial logistic regression model was adjusted for nutritional status, like the Body Mass Index (BMI), to minimize this bias. This information is described on Page 4 Line 178-180, and Table 2. Furthermore, some studies with head and neck cancer risk and factors associates also did not assess socioeconomic status, only BMI and others covariates (DOI: 10.1371/journal.pone.0220067; DOI: 10.3945/ajcn.112.044859; DOI: 10.1093/aje/kwr468; DOI: 10.1097/EDE.0000000000000902).

On smoking habits, which occurred in the majority of individuals in this study and is one of the most important risk factors in HNSCC, we included the smoking status as a covariate in the multinomial logistic regression model to minimize biases. This information is described on Page 4 Line 178-180, and Table 2.

The authors respectfully thank you for all your comments and suggestion. The contributions were fundamental for the improvement of the manuscript.